# Efficacy and Safety of Acupuncture and Related Techniques in the Management of Oncological Children and Adolescent Patients: A Systematic Review

**DOI:** 10.3390/cancers16183197

**Published:** 2024-09-19

**Authors:** Esther Martínez García, M. Betina Nishishinya Aquino, Ofelia Cruz Martínez, Yiming Ren, Ruyu Xia, Yutong Fei, Carles Fernández-Jané

**Affiliations:** 1Integrative Pediatric Oncology Unit, Pediatric Cancer Center Barcelona, Sant Joan de Déu Children’s Hospital, 08950 Barcelona, Spain; esther.martinez@sjd.es (E.M.G.); mbetina.nishishinya@sjd.es (M.B.N.A.); 2Pediatric Neuro-Oncology, Pediatric Cancer Center Barcelona, Sant Joan de Déu Children’s Hospital, 08950 Barcelona, Spain; ofelia.cruz@sjd.es; 3Graduate School, Beijing University of Chinese Medicine, Beijing 100029, China; 20230941408@bucm.edu.cn; 4Center for Evidence-Based Chinese Medicine, Beijing University of Chinese Medicine, Beijing 100029, China; xiary@bucm.edu.cn (R.X.); feiyt@bucm.edu.cn (Y.F.); 5Tecnocampus, Universitat Pompeu Fabra, Mataró-Maresme, 08302 Barcelona, Spain

**Keywords:** systematic review, children, oncology, cancer, acupuncture, patient safety

## Abstract

**Simple Summary:**

Acupuncture is becoming a popular complementary treatment in pediatric cancer care for managing symptoms like nausea and vomiting that result from cancer treatments. This review aims to evaluate how effective and safe acupuncture and similar techniques are for children with cancer. By analyzing data from multiple studies, the authors found that acupuncture may help reduce nausea and vomiting in young cancer patients. However, there is not enough evidence to make conclusions about its effects on number of vomiting episodes, need for anti-nausea medication, or fatigue. More high-quality research is needed to confirm these benefits and ensure the safety of acupuncture for children undergoing cancer treatment.

**Abstract:**

*Background*/*Objectives*: Oncology acupuncture is emerging as a complementary treatment in pediatric cancer care centers. It is valued for its potential to manage symptoms associated with cancer and treatment toxicities without increasing polypharmacy. The aim of this review is to evaluate the efficacy and safety of acupuncture and related techniques in alleviating symptoms of cancer treatment in pediatric oncology patients. *Methods*: A comprehensive search was conducted across nine databases, including PubMed, Cochrane Library, and CNKI, up to June 2023. Inclusion criteria focused on randomized and quasi-randomized controlled trials involving pediatric oncology patients undergoing needle acupuncture or related techniques. Study selection and data extraction were independently performed by pairs of authors. Results were analyzed narratively, and meta-analysis was performed when possible. *Results*: Results suggest that acupuncture may help manage symptoms such as nausea and vomiting in pediatric oncology patients. However, the quality of evidence was generally low, and further research is required to substantiate these findings. *Conclusions*: Acupuncture shows promising results as a complementary treatment for reducing nausea and vomiting in pediatric oncology. However, current evidence is insufficient to draw conclusions for other outcomes, such as the number of vomiting episodes, reduction in antiemetic medication use, or fatigue. High-quality, rigorously designed studies are necessary to better understand the clinical relevance and safety of acupuncture in this vulnerable population.

## 1. Introduction

Among complementary treatments in cancer care, acupuncture is the one that has shown the highest level of evidence, and even the WHO (World Health Organization), in 1979, recognized its efficacy and safety [1]. The simplicity of the procedure, the low cost, the null pharmacological interaction, the high safety profile, and its potential clinical benefit, make this complementary treatment the object of much clinical interest and many research projects.

Acupuncture encompasses a variety of techniques beyond the widely known filiform needle acupuncture. Additional commonly practiced techniques include moxibustion, which involves applying heat to acupuncture points, electroacupuncture, which utilizes electrical current through inserted needles, laserpuncture using a light stimulus, and ear acupuncture, focusing on specific points on the ear. Acupressure, involving digital pressure on acupuncture points, is also commonly employed. These techniques expand the range of options available in acupuncture, allowing for personalized treatment approaches. It is worth noting that these techniques are just a few examples, as acupuncture continues to evolve, with practitioners utilizing other specialized techniques and variations to address individual needs.

Acupuncture is a medical technique widely used in leading cancer centers around the world as a complementary treatment in oncology, because it does not increase polypharmacy, presents a large margin of safety in expert hands [2,3,4], and does not interfere with the patient’s usual treatment. The recently published ASCO guidelines suggest acupuncture for the management of cancer and musculoskeletal pain [5]. It has shown efficacy for cancer pain, or arthralgias secondary to aromatase inhibitors [6]. Oncological acupuncture offers potential benefits for managing symptoms and adverse effects related to cancer and its treatments. For instance, it has been shown to provide relief from chemotherapy-induced nausea and vomiting, reduce pain, and improve overall quality of life for cancer patients. These benefits are especially significant given that current standard treatments may not always be fully effective or could lead to increased polypharmacy and associated toxicity [7].

In pediatric oncology, traditional and complementary medicine is widely used [8]. In a systematic review, the prevalence of the use of traditional Chinese medicine in pediatric oncology ranged from 6 to 100% [9].

Most of these data are related to adult cancer acupuncture. There are still many knowledge gaps in oncology acupuncture and pediatric oncology requiring a comprehensive review of the literature to evaluate the existing scientific evidence and detect potential uses of the technique in clinical applications commonly used in pediatric cancer patients, beyond the safety and feasibility of the procedure.

The ASCO guidelines analyze the evidence on the different interventions for the management of cancer pain in adults, but the evidence is not conclusive enough to recommend in children with cancer [5].

Having safe and effective techniques and treatments that do not increase polypharmacy, that reduce the toxicity of routine cancer treatment, and improve the patient’s quality of life is of great interest in clinical practice. That is why a review of the literature is necessary to allow us to analyze whether acupuncture and its related techniques can be a complementary treatment of interest for our patients. The aim of this review is to assess the evidence on the efficacy and safety of acupuncture and its related techniques compared with no treatment, usual care, or sham interventions in symptoms associated with cancer and toxicities secondary to cancer treatment in pediatric oncology patients.

## 2. Materials and Methods

The protocol of this review was registered in PROSPERO (the International Prospective Register of Systematic Reviews, CRD42022330790), and the Preferred Reporting Items for Systematic Reviews and Meta-Analyses (PRISMA) and its extension for acupuncture (PRISMA-A) were used to write the manuscript.

### 2.1. Information Sources

Comprehensive search strategies were developed and executed on 9 Chinese and Western databases. The search was run in June 2024 in the following databases: Pubmed, Cochrane library, Pedro, Cinhal, Embase, CNKI, Wanfang, VIP, and Sinomed. Searching was not limited by language. Reference lists of review articles and included studies were searched for additional studies.

We used specific search equations adapted for each database, and the following terms were used in English databases: “Acupuncture therapy”, “Moxibustion”, “Electroacupuncture”, “Ear acupuncture”, “Press task needle”, “Pediatrics”, “Children”, “Adolescent”, “Neoplasms”, “Cancer”, “Oncology”, “Tumor”, “Sarcoma”, “Leukemia”, “Chemotherapy”, “Radiotherapy”, “Immunotherapy”, “Randomized controlled trials”. Full search strategies for each database can be seen in Appendix A.

### 2.2. Study Selection

Study selection was performed using the Rayyan tool for systematic reviews. After eliminating duplicates, authors (EM, BN, CF, YF, RX, YR) by pairs independently decided the adequacy of the results according to the inclusion criteria, first by the titles and abstracts and then using the full text. In case of discrepancies, a consensus between the two reviewers was reached through discussion or, in case it was not possible, by a third reviewer. A pilot selection process was performed to ensure consistency.

Studies were included if they (1) included randomized clinical trials (RCTs) or quasi-randomized controlled trials evaluating the effects of acupuncture techniques in pediatric oncology participants and showed original data published in a peer-reviewed journal; (2) included children from birth to 18 years of age, inclusive, with any condition and stage of cancer; (3) included needle acupuncture or related techniques (including wire acupuncture, electroacupuncture, transcutaneous electrical neve stimulation (TENS), moxibustion, acupressure, auricular acupuncture, acupressure, laser acupuncture, embedded needles, or pharmacological modalities such as acupoint injection) alone or in combination with regular treatment; (4) included no treatment, usual care, or sham intervention as control; (5) included at least one of the following outcomes: nausea and vomiting, diarrhea or constipation, pain, fatigue, quality of life, quality of sleep, emotional state (anxiety and depression), or safety.

Trials referring to techniques such as dry needling or combining acupuncture with non-conventional interventions (such as herbal remedies) were excluded.

### 2.3. Data Extraction

Data were extracted independently by pairs (EMG, BNA, OCM, CF-J, YF, RX, YR) using extraction forms specifically designed for this review, which included information on the publication, participants, intervention and control groups, outcomes, results, and conclusions. The form also included the items of the Cochrane Risk of Bias Assessment tool. Discrepancies were solved by a third reviewer. A pilot extraction was performed to verify the usability and understanding of the extraction form using one article. In case of missing information, authors were contacted by email.

### 2.4. Publication Bias

We initially planned to assess the risk of publication bias by generating a funnel plot, which would allow for the visual detection of asymmetries in the distribution of effect sizes. This method is generally recommended when the meta-analysis includes at least 10 studies, as fewer studies may lead to unreliable or misleading interpretations. However, due to the small number of studies included in our analysis, it was not possible to construct a meaningful funnel plot, and as such, a formal assessment of publication bias could not be performed.

### 2.5. Data Synthesis

Due to the heterogeneity of the studies and the data obtained, it was only possible to perform 2 meta-analyses. For this reason, we used a narrative approach for most of the results according to each control type, study outcome, and intervention type.

For the meta-analysis, we used RevMan software (version 5.4.1). A mean difference (MD) was used to combine studies with quantitative outcomes that used the same units, while a standardized mean difference (SMD) was employed for studies using different measurement scales. Also, a random effects model was used to calculate the 95% confidence intervals (95%CI) due to the diversity of the techniques and treatment protocols used. To assess the heterogeneity of the meta-analysis, we used the I^2^ statistic. Heterogeneity was considered low for values <30%, moderate for values between 30% and 70%, and high for values >70%. A sensitivity analysis was conducted to assess the robustness of the results by recalculating the results, systematically excluding each trial one at a time.

### 2.6. Summary of Findings and Evidence Assessment

We assessed the certainty of the results in a summary-of-findings table, using the GRADEpro tool. However, this was limited by the fact that we could only perform two meta-analyses.

## 3. Results

### 3.1. Selection Process

A total of 723 results were obtained in the electronic database search. After removing duplicates, a total of 594 results were retrieved by title and abstract and 45 by the full text. Finally, 11 trials were included in this review. Details of the selection process are shown in Figure 1, and reasons for the excluded results are detailed in Appendix A.

### 3.2. Study Description

Included studies were performed in Germany, Canada, the United States, Brazil, Taiwan, China, Turkey, Indonesia, and Iran, between 2008 and 2024.

Characteristics of the included studies are shown in Table 1 and Table 2.

#### 3.2.1. Study Design

We identified three multicentric [13,14,15] and eight unicentric trials [10,11,12,16,17,18,19,20], which included four parallel [12,13,14,18] and five crossover randomized designs [10,15,16,17,20]. Two of the trials were pilot studies, and their primary objective was to assess the feasibility of a larger trial [16,20].

#### 3.2.2. Participants

The sample sizes across the studies varied significantly. Four large studies had samples ranging from 96 to 187 participants [13,14,17,19]. Three other studies had more moderate sample sizes, with between 40 and 74 participants each [10,11,12]. The remaining four studies were much smaller, and only included from 10 to 23 participants [15,16,18,20]. The mean age of participants across the studies ranged from 4.75 to 13.6 years, and the proportion of male children was from 41 to 68.9%; however, one study did not report gender distribution [17]. The neoplastic pathologies covered in these studies were diverse, including osteosarcoma, Ewing’s sarcoma, Hodgkin’s lymphoma, acute lymphoblastic leukaemia, rhabdomyosarcoma, medulloblastoma, and liver cancer, among others.

#### 3.2.3. Interventions

All the included studies analyzed the effect of acupuncture techniques combined with regular antiemetic treatment.

Acupressure was utilized in five studies, with one study incorporating two intervention modalities and another combining acupressure with ear acupuncture. Acupressure bands were applied in three trials on bilateral wrists at P6 [10,13,16]. In all studies, the bands were applied before the initiation of the chemotherapy and were maintained until discharge [16], over several days [13] or for 15 min [10]. Manual acupressure was used in three other studies on P6 [10,19] or both P6 and ST36 [14]. Two of the studies used a single session before chemotherapy initiation [10] or on the second day of chemotherapy [14], with bilateral finger stimulation of each point for two to three minutes. In the third study, alongside the manual stimulation of P6, ear acupuncture at Shenmen and stomach was also used with bilateral ear seeds [19]. Points were stimulated three to four times a day until the competition of the chemotherapy treatment.

Filiform needle acupuncture was used in one study [15]. Acupuncture was applied on day 1, before starting chemotherapy, and was offered on consecutive days of the chemotherapy course at patient’s demand. Needles were placed unilaterally or bilaterally for 20–45 min, and needles were inserted and manipulated until achieving a “De Qi” sensation. Point combinations depended on the acupuncturist’s decision. Most used points were P6, ST36, CV12, and LI4.

Press needles were used in two studies [11,12]. In the first trial, needles were inserted in P6 and ST36 before the chemotherapy session, and manually stimulated for 60 s three times a day until the third day after chemotherapy [12]. In the second one, subcutaneous press needles were inserted bilaterally at ST36, P6, and LI4 for 48 h. The press needles were manually stimulated for 1 min at 1 h before, at the beginning of, every 1 h during, and 1 h after the chemotherapy [11].

Laser acupuncture was assessed in one study [18], using a visible Class II red laser (Ibramed^®^). The treatment was applied using variable and continuous stimulation frequencies, a wavelength of 660 nm, and a power density of 30 mw and 3 joules, at days 1, 2, 3, and 5 of chemotherapy. Laser acupuncture took place on the first day of each cycle for a total of 26 cycles at P6, LI4, SP6, ST36, BL20, CV10, and CV12.

Ear acupuncture alone was included in two studies [17,20], both using pressure stimulation with ear patches during 4 or 7 days. Both studies used six points, which included Shenmen, stomach (CO4), and other digestive points.

#### 3.2.4. Control

Regarding the control treatments, seven studies [10,12,13,14,16,18,20] used a sham intervention which included sham acupressure bands and sham laser acupuncture at the same acupuncture points as the intervention group, and acupressure performed at acupuncture points not related to CINV. In four studies, the control group received only conventional antiemetic therapy [11,15,17,19].

#### 3.2.5. Outcomes

We identified four study outcomes, which included nausea and vomiting, use of antiemetic rescue medication use, fatigue, and adverse events.

Nausea and vomiting were the most common outcome, being measured in eleven trials. Evaluation tools for this outcome were very diverse and included The 24 h Nausea and Vomiting Episode Number and Severity Assessment Form, the Pediatric Nausea Assessment Tool, daily number of emetic episodes, the Adapted Rhodes Index of Nausea and Vomiting (ARINV) and the ARINV for Pediatrics by Child (ARINVc), modified methods of Morrow questionnaires, intensity of nausea and number of vomiting episodes scale of the National Cancer Institute, the WHO criteria, the NCI Common Terminology Criteria for Adverse Events v3.0, and the Functional Living Index-Emesis (FLIE) scale. Also, the use of antiemetic rescue medication was assessed in two studies [12,15].

Fatigue was assessed using the Fatigue Scale Child (FSC) in one study [14], and adverse events were assessed in only four trials [12,13,15,16].

### 3.3. Risk of Bias

The risk of bias assessment for all included studies is summarized in Table 3. An expanded table with the rationale for each assessment is available in the Appendix A. Overall, studies were considered to have an unclear or high risk of bias.

Regarding selection bias, only one study had low risk of bias both on random sequence generation and allocation concealment [15], while six studies were considered to have an unclear risk of bias on allocation concealment [11,12,13,14,19,20], and three on both randomization and allocation concealment [16,17,18]. One trial was rated as having a high risk of bias for the randomization sequence because, although children were randomized into two groups (manual/wristband acupressure), in both groups children were first given the real treatment and sham intervention was given in the second chemotherapy cycle [10].

Regarding performance bias, due to the nature of the interventions, personnel were not blinded in any intervention, while participants were considered blinded in seven trials due to the use of a sham intervention [10,12,13,14,15,16,20].

For detection bias, outcome assessment was considered unclear in six studies due to lack of information provided [11,14,15,16,17,19]. For attrition bias, two studies were considered at high risk of bias due to a low participant completion rate [16,20], and two studies did not provide enough information to assess this item [12,18]. Finally, regarding reporting bias, only two trials provided the required information to obtain the study protocol [12,13], while it was unavailable in the other ten studies.

### 3.4. Efficacy of the Interventions

#### 3.4.1. Intervention vs. Sham Intervention

##### Nausea and Vomiting

Five trials assessed the effect of acupressure on nausea and vomiting. Altuntaş et al. reported a reduction in the number and severity of nausea and vomiting episodes in both manual and wristband interventions, compared with sham, 24 h after the intervention (*p* < 0.05) [10]. Ghezelbash et al. found a statistically significant reduction in the intensity of nausea and vomiting immediately (*p* = 0.02) and one hour after the intervention (*p* ≤ 0.001), but not after 12 h [14]. Conversely, Dupuis et al. and Jones et al. did not find any reduction in nausea severity or the number of vomiting episodes during treatment, nor in daily vomiting using acupressure bands, immediately after the intervention [16], nor at 24 h, or 1, 2, 3, or 4 days post-intervention [13].

One trial used press needles for nausea and vomiting [12]. Bintoro et al. found statistically significant reductions in the RINVR scores on the chemotherapy day (*p* = 0.001) and the 3rd day post-chemotherapy (*p* = 0.005), but not on the 6th day post-chemotherapy (*p* = 0.213).

Regarding the use of ear acupuncture, Yeh et al. reported no statistical differences on occurrence, severity, or duration of nausea and vomiting [20].

For laser acupuncture, Varejão et al. reported a lower level of nausea intensity and fewer vomiting episodes on days 2 and 3 (*p* = 0.001), but not on days 1, 4, and 5 [18].

Two meta-analyses were performed for this outcome, one regarding the severity of nausea and the other for the number of vomiting episodes.

For the severity of nausea after the intervention, trials from Altuntaş [10], including both manual and acupressure bands, Ghezelbasch [14], and Jones [16] were plotted, with a total of 244 participants. Results showed a statistically significant reduction in nausea after the chemotherapy session (SMD: −0.57; 95%CI: −0.83, −0.31; *p* = 0.0001, I^2^ = 0%) (Figure 2a).

Regarding the number of vomiting episodes, meta-analysis included results from Altuntaş [10] and Jones [16], with a total of 124 participants. Results did not show a statistical improvement on this outcome (MD: −0.16; 95%CI: −0.35, 0.03; *p* = 0.09; I^2^ = 0%) (Figure 2b).

The sensitivity analysis of both meta-analyses showed robustness of the results (Appendix A).

##### Antiemetic Rescue Medication Use

Only Altuntaş et al. assessed this outcome and did not observe a reduction in the consumption of additional antiemetic drug use [10].

##### Fatigue

Only Ghezelbasch et al. assessed fatigue using acupressure. They found a statistical improvement in fatigue intensity 1 h post-intervention (*p* ≤ 0.01) but not at 12 h post-intervention [14].

#### 3.4.2. Interventions Plus Regular Antiemetics vs. Antiemetic Alone

##### Nausea and Vomiting

One trial assessed the effect of filiform needle acupuncture. Gottschling et al. reported a statistically significant improvement in episodes of retching and vomiting. However, mean differences and 95% CI between groups were not provided [15].

Liu et al. assessed the efficacy of ear acupuncture and found a statistically significant improvement in degree of vomiting control before and after the crossover (*p* = 0.016, *p* = 0.007) [17].

Xie et al. assessed the use of acupressure combined with ear acupuncture, and the severity of nausea and the number of vomiting episodes showed statistically significant improvement in days 2~7 (*p* < 0.01), but not on the first day (*p* = 0.083) [19].

Bai et al. assessed the effect of subcutaneous embedding acupuncture, and the severity of nausea and vomiting showed statistically significant reduction on the first (*p* < 0.05) and second days (*p* = 0.01), while there were no differences using the FLIE (*p* < 0.05) [11].

Due to the heterogeneity of the outcome nature (dichotomous, categorical, and continuous) and endpoints, meta-analysis of the studies was considered not appropriate.

##### Antiemetic Rescue Medication Use

Only Gottschling et al. assessed the use of antiemetic rescue medication. They reported a reduction for phenothiazine (*p* = 0.001) using filiform needle acupuncture but not dexamethasone [15].

#### 3.4.3. Adverse Events

For acupressure bands, discomfort due to the tightness of the bands was reported by four out of eighteen participants in the trial of Jones et al. [16], and twelve (six for the real band and six for the sham band) out of one hundred eighty-seven in the study of Dupuis et al. [13].

Regarding filiform needle acupuncture, the study of Gottschling et al. reported that four out of twenty-three participants experienced pain from needling, but all adverse effects were minor and transient [15].

For the use of ear acupuncture, Yeh et al. reported that three out of ten participants took off the tapes due to itching [20].

### 3.5. Summary of Findings and Evidence Assessment

A summary of findings can be seen in Table 4.

## 4. Discussion

This study aimed to evaluate the effectiveness and safety of various acupuncture techniques in managing symptoms related to oncological treatments in children. The systematic review identified twelve trials, incorporating interventions such as acupressure, press needle, filiform needle, laser acupuncture, and ear acupuncture.

The analysis revealed heterogeneous results regarding the efficacy of acupuncture in alleviating nausea and vomiting. However, meta-analysis showed statistical benefits of acupuncture techniques compared with sham, in reducing nausea after the intervention but not the number of vomiting episodes. Regarding other outcomes, one trial [15] indicated that acupressure could improve fatigue 1 h post-intervention compared to a sham intervention, but the benefit did not persist at 12 h. For the reduction of antiemetic medication, one trial reported no statistical differences comparing acupressure with sham [16], while another concluded that adding filiform needle acupuncture to regular treatment reduced the need for phenothiazine but did not affect the requirement for dexamethasone [15].

The variability in the results of the included trials can be attributed to several factors. First, the small size of some trials, especially the ones with fewer than 25 participants [15,16,18,20], limits their statistical power and might have prevented detection of existing differences between groups. Second, differences could also be attributed to the diverse methodologies employed across the trials, including different acupuncture techniques, control groups, assessment methods, and endpoints.

Safety assessments showed that minor and transient adverse effects were associated with acupuncture. Local pain was reported by 17.4% of patients in one study involving needle acupuncture and one patient using press needles, discomfort from tight bands was noted by 22% of patients in an acupressure study, and auricular itching led three patients to remove their tapes in an ear acupuncture study. Despite these adverse effects, acupuncture techniques were generally well-tolerated, with no serious adverse events reported. These adverse effects, although minor, should be carefully considered in the clinical application of acupuncture in pediatric oncology.

A significant limitation of the included trials was the lack of usable data to perform a comprehensive meta-analysis. Even when this was possible, such as in the comparison of acupuncture techniques with sham in reducing nausea, only three of the twelve studies could be included. One reason for this was that some studies assessed nausea severity using a categorical approach (no nausea/mild/moderate/severe) [13] or had such small samples that data were reported as medians and ranges [12]. Additionally, some studies did not provide mean differences or risk/odds ratios between groups, relying solely on *p*-values to indicate statistical significance. This practice hinders the evaluation of the clinical relevance of the findings, as *p*-values alone do not convey the magnitude or direction of the effects observed. Furthermore, the methodological quality of the studies was not consistently reported, and the small number of trials precluded the assessment of publication bias. These limitations in performing a meta-analysis further restricted the ability to assess the quality of the evidence using the GRADE approach.

Despite these limitations, this study has several strengths. The use of a robust systematic methodology and a highly sensitive search strategy, including searches of Chinese-language databases, ensured comprehensive coverage of the relevant literature. This thorough approach enhances the reliability of the findings and provides a solid foundation for future research.

The present study aligns with the findings of Dana C. Mora et al. [21], who conducted a systematic review and meta-analysis to evaluate the efficacy of complementary and alternative medicine modalities in treating adverse effects of anti-cancer treatments among children and young adults. In their review, Mora et al. found that acupuncture significantly reduced the intensity and frequency of CINV compared to controls (SMD: 0.59; 95%CI: −0.85, −0.33), which is consistent with our results. However, there are some differences between the two studies. First, our study has an updated search period until June 2024, while Mora et al. only included studies until April 2021; this allowed us to include four more acupuncture trials published during 2022 [10,11,12,19]. Second, in the subgroup analysis where Mora et al. analyzed the effect of acupuncture, they plotted together different comparisons, while we separated studies using sham and studies adding acupuncture to usual treatment compared only with usual treatment.

## 5. Conclusions

Based on current evidence, acupuncture may provide some benefits in managing chemotherapy-induced nausea (CIN) related to oncological treatments in children when compared to a sham intervention. However, its impact on other outcomes, such as reducing vomiting episodes, alleviating fatigue, and decreasing the need for rescue medication, remains unclear. The heterogeneity of the existing trials, as well as their methodological and reporting limitations, highlight the need for further high-quality research in this field. Future studies should aim to standardize outcome measures, improve reporting practices, and include larger sample sizes to better assess the clinical relevance and safety of acupuncture interventions in this patient population. This will help to establish more definitive conclusions and potentially guide clinical practice in managing chemotherapy-induced nausea and other related symptoms in children undergoing oncological treatments.

## Figures and Tables

**Figure 1 cancers-16-03197-f001:**
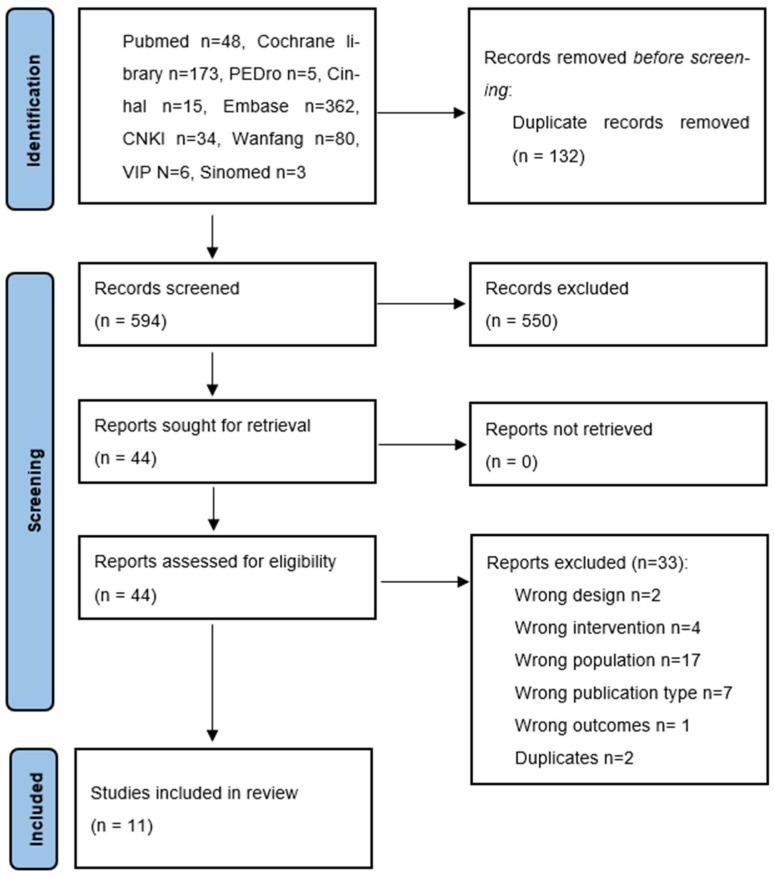
PRISMA Flow Diagram.

**Figure 2 cancers-16-03197-f002:**
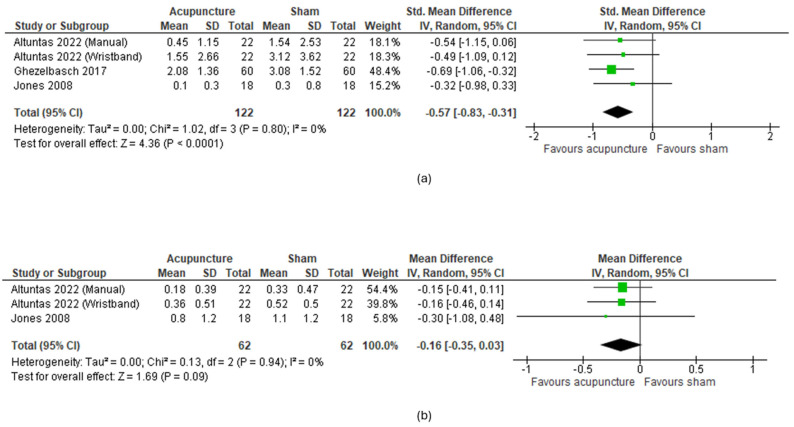
**Meta-analysis of acupuncture vs. sham for nausea and vomiting**. (**a**) Effect on nausea severity; (**b**) effect on number of vomiting episodes.

**Table 1 cancers-16-03197-t001:** Study characteristics.

**Author/Year** **Country**	**Study Design**	**Setting and Funding Source**	**Participants**	**Outcomes (Measurement Instruments)**
**Altuntaş 2022** Turkey [10]	Crossover RCT Single-center	Pediatric oncology clinic **Funding:** Coordination Unit of Akdeniz University	**Total participants:** 46 (M:27, F:17) **Mean age:** 10 ± 4.13 **Cancer diagnosis:** (not reported) **Other:** Not receiving chemotherapy for the first time.	**Primary** CIN/CIV (24 h Nausea and Vomiting Episode Number and Severity Assessment Form) **Secondary** Nonroutine antiemetic drug use (recording table)
**Bai 2024** China [11]	Parallel RCT Single-center	Beijing Children’s Hospital Affiliated to Capital Medical University **Funding** Beijing Traditional Chinese Medicine Administration project and Beijing Children’s Hospital nursing special project	**Total participants:** 74 (M:51; F:23) **Mean age:** IG: 9.95 ± 2.19, CG: 9.92 ± 2.49) **Cancer diagnosis:** Soft tissue sarcoma (rhabdomyosarcoma, 29; Non-rhabdomyosarcoma, 24; Ewing’s sarcoma 21) **Other:** Receiving Vincristine + Doxorubicin + Cyclophosphamide. No history of gastrointestinal disease. There was no nausea or vomiting before chemotherapy. Estimated survival time >6 months.	**Primary** CIN/CIV (WHO criteria, FLIE Scale)
**Bintoro 2022** Indonesia [12]	Parallel RCT Single-center	Integrated inpatient service installation **Funding:** No financial assistance received	**Total participants:** 60 (M:25, F:35) **Mean age:** IG: 12.10 ± 3.06, CG: 10.83 ± 3.72 **Cancer diagnosis:** Leukemia, lymphoma, osteosarcoma, germ cell tumor, central nervous system tumor, rhabdomyosarcoma and liver tumor **Other:** platelet values ≥20,000/µ and neutrophil values ≥1000/µ	**Primary** CIN/CIV (RINVR) **Secondary** Adverse event (patient reporting)
**Dupuis 2018** Canada USA [13]	Parallel RCT Multicenter	27 institutions **Funding** National Cancer Institute, the Children’s Oncology Group and the SunCoast CCOP Research Base	**Total participants:** 187 (M:96; F:69) **Mean age:** 12.7 ± 4.2 **Cancer diagnosis:** Osteosarcoma, Ewing sarcoma, Hodgkin lymphoma… **Other:** Non-relapsed cancer, receiving cisplatin 50 mg/m^2^ per dose.	**Primary** CIN/CIV (PeNAT) Daily number of emetic episodes **Secondary** Adverse event (patient reporting)
**Ghezelbasch** **2017** Iran [14]	Parallel RCT Multicenter	Oncology unit in two pediatric educational hospitals in Tehran **Funding** Tehran University of Medical Sciences and Health Services	**Total participants:** 120 (M:82; F:38) **Mean age:** 9.98 ± 1.55 **Cancer diagnosis:** Acute lymphoblastic leukemia **Other:** No prior experience of chemotherapy or acupressure, more than three months anticipated survival.	**Primary** CIN/CIV (ARINVc) Fatigue (FSC)
**Gottschling** **2008** Germany [15]	Crossover RCT Multicenter	German pediatric oncology centers (Berlin, Bonn, Erlangen, Hannover, and Homburg) **Funding** C.D. Foundation and the Friedrich-Spicker Foundation	**Total participants:** 23 (M:10; F:13) **Mean age:** 13.6 ± 2.9 **Cancer diagnosis:** Ewing sarcoma, rhabdomyosarcoma, osteosarcoma, undifferentiated sarcoma, synovial sarcoma **Other:** Scheduled to receive at least three identical, consecutive courses of highly emetogenic chemotherapy.	**Primary** Antiemetic rescue medication **Secondary** Number of episodes of retching and vomiting (short open-form essay)
**Jones 2008** USA [16]	Pilot Crossover RCT Single-center	The pédiatrie oncology patients at Brenner Children’s Hospital (Winston-Salem, NC) **Funding** No data available	**Total participants:** 18 (M:9; F:9) Range age: 5–19 **Cancer diagnosis:** Rhabdomyosarcoma, Ewing, medulloblastoma, osteosarcoma… **Other:** Received emetogenic chemotherapy agents, an antitumor antibiotic, or high-dose cytarabine.	**Primary** CIN/CIV (MMQ) **Secondary** Previous knowledge and experience with acupressure or acupuncture Expectations of nausea prevention Episodes of emesis (presence and degree) Side effects Satisfaction
**Liu 2018** China [17]	Crossover RCT Single-center	Children’s Hospital Affiliated to Soochow University **Funding** Suzhou Science, Education and Health project	**Total participants:** 96 (M:-; F:-) **Mean age:** 4.75 **Cancer diagnosis:** Acute lymphoblastic leukemia **Other:** Receiving MTX 2 g/m^2^ per dose and 6-MP 25 mg/m^2^ per dose.	**Primary** Vomiting (WHO criteria)
**Varejão 2019** Brazil [18]	Parallel RCT Single-center	Instituto Nacional de Câncer, Rio Janeiro **Funding** No financial assistance received	**Total participants:** 17 (M:9; F:8) **Mean age:** GA 12.57 ± 2.57, GB 14.7 ± 2.11 **Cancer diagnosis:** Osteosarcoma, rhabdomyosarcoma, Ewing’s sarcoma **Other:** ---	**Primary** CIN/CIV intensity (Scale of the National Cancer Institute)
**Xie 2016** China [19]	Parallel RCT Single-center	Children’s Hospital Affiliated with Soochow University **Funding** Suzhou Science, Education, and Health project	**Total participants:** 104 (M:65; F:39) **Mean age:** 8.13 ± 2.45 **Cancer diagnosis:** Acute lymphoblastic leukemia **Other:** ---	**Primary** CIN/CIV (NCI-CTCAEV 3.0 Criteria)
**Yeh 2012** Taiwan [20]	Pilot Crossover RCT Single-center	Children’s hospital in Taiwan **Funding** Not reported	**Total participants:** 10 (M:6; F:4) **Mean Age** 13.29 ± 3.31 **Cancer diagnosis:** Leukemia and other solid tumors **Other:** ---	**Primary** CIN/CIV (MANEQ)

IG: Intervention group, CG: control group, M: male, F: female, ALL: acute lymphoblastic leukemia, ARINVc: Adapted Rhodes Index of Nausea and Vomiting for Pediatrics by Child, CIN: chemotherapy-induced nausea, CIV: chemotherapy-induced vomiting, CRF: cancer-related fatigue, FLIE: Functional Living Index-Emesis, FSC: Fatigue Scale Child, MANEQ: Morrow Assessment of Nausea and Emetics, a self-reported questionnaire (adapted for children), MMQ: modified methods of Morrow questionnaires, NCI-CTCAEV: National Cancer Institute Common Terminology Criteria for Adverse Events; PeNAT: Pediatric Nausea Assessment Tool, RINVR: Rhodes Index of Nausea, Vomiting, and Retching.

**Table 2 cancers-16-03197-t002:** Intervention and control description.

**Author** **Year**	**Intervention**	**Control**
**Altuntaş 2022** [10]	**Acupressure** Stimulation type: I1: Manual stimulation with fingers I2: Wristband (Sea-Bands^®^) Points: P6 Treatment start: 30 min before chemotherapy Regime and duration: I1: 2 min stimulation, I2: 15 min stimulation **Antiemetic therapy:** (not reported)	**Placebo Manual Acupressure.** Same steps, but pressure was applied in a nonactive way without applying sufficient pressure **Placebo Wristband** Same steps, but removing the stud of the acupressure wristband
**Bai** **2024** [11]	**Embedding acupuncture** Stimulation type: Vertically inserting needle and pressing the back film to make it stick firmly, applying embedding acupuncture (0.5 cm depth) until the sensation of local acid swelling, and retention 1 min for each point Points: Bilateral ST36, P6, LI4 Treatment start: 1 h before, at the beginning of, every 1 h during, and 1 h after the infusion of chemotherapy drugs Regime and duration: Until the completion of chemotherapy (at least 48 h) **Antiemetic therapy:** Ondansetron	**Antiemetic therapy**
**Bintoro 2022** [12]	**Press needles** Stimulation type: After the press needle was inserted, 60 s massages were performed, stimulation was repeated three times a day plus every time the patient felt nauseous or wanted to vomit. Points: P6 and ST36 Treatment start: Before chemotherapy Regime and duration: Every day, until third day after chemotherapy **Antiemetic therapy:** (not reported)	**Sham intervention** Plaster Plesterin^®^ without needles at the same location and duration as the treatment group without any stimulation **Antiemetic therapy**
**Dupuis** **2018** [13]	**Acupressure** Stimulation type: Bilateral acupressure wristbands (Sea-Bands^®^) Points: P6 Treatment start: 30 min prior to the administration of the first chemotherapy dose Regime and duration: Continuous stimulation during the acute and delayed phase; participants could remove the bands four times a day for 15 min **Antiemetic therapy:** Granisetron or ondansetron plus dexamethasone or aprepitant	**Sham intervention:** Bands with no internal plastic stud **Antiemetic therapy**
**Ghezelbasch** **2017** [14]	**Acupressure** Stimulation type: 3 min per point bilateral finger acupressure Points: P6 and ST36 Treatment start: On the second day of chemotherapy Regime and duration: Single treatment **Antiemetic therapy**: (No data available)	**Sham intervention:** Same stimulation at points SI3 and LI1 **Antiemetic therapy**
**Gottschling** **2008** [15]	**Filiform needle** Stimulation type: Bilateral needle insertion and manipulation until the achievement of “De Qi” and retention for 20–45 min Points: Individualized point combination, most used points were P6, ST36, CV12, LI4 Treatment start: On day 1, before starting chemotherapy Regime and duration: At consecutive days of the chemotherapy course on patient’s demand **Antiemetic therapy** Ondansetron or tropisetron, additional dexamethasone or phenothiazines	**Antiemetic therapy**
**Jones** **2008** [16]	**Acupressure** Stimulation type: Bilateral acupressure wristbands (Sea-Bands^®^) Points: P6 Treatment start: Prior to the initiation of chemotherapy (from minutes to hours) Regime and duration: Bands were worn until the completion of chemotherapy (usually until the time of discharge). **Antiemetic therapy** Ondansetron or granisetron and dexamethasone, others	**Sham intervention:** Bands with no internal plastic stud **Antiemetic therapy**
**Liu** **2018** [17]	**Ear acupuncture** Stimulation type: Bilateral ear seeds with adhesive tape, ear pressing point by the patient Points: Shenmen (TF4), stomach (CO4), liver (CO12), spleen (CO13) Treatment start: Prior to the initiation of chemotherapy (30 min) Regime and duration: Four to six times a day, each point retention for 20–30 s, stopped for a few minutes and repeated the pressing three to five times, intermittently for 4 days until the end of observation **Antiemetic therapy** Ondansetron before chemotherapy and 3 days after	**Antiemetic therapy**
**Varejão 2019** [18]	**Laser acupuncture** Stimulation type: Visible Class II red laser (Ibramed^®^) variable and continuous stimulation frequencies, a wavelength of 660 nm, and a power density of 30 mw and 3 joules; 1 min for each point, totaling 6 min Points: P6, LI4, SP6, ST36, BL20, CV10, CV12 Treatment start: Minutes before the start of the first day of chemotherapy Regime and duration: On the first day of each chemotherapy cycle for 26 cycles **Antiemetic therapy** Ondansetron and dexamethasone before chemotherapy	**Sham intervention** Sham laser at the same points
**Xie** **2016** [19]	**Acupressure and Ear acupuncture** Stimulation type: 3–5 min per point bilateral finger acupressure; bilateral ear seeds with adhesive tape and pressing stimulation was applied for 1 min per point until feeling mild distension, tingling, heat Points: Acupuncture points: Bilateral P6 Ear acupuncture points: Shenmen (TF4), stomach (CO4) Treatment start: On the first day of chemotherapy Regime and duration: Three times a day for acupressure and four times a day for ear acupuncture until the completion of chemotherapy **Antiemetic therapy:** Ondansetron	**Antiemetic therapy**
**Yeh** **2012** [20]	**Ear acupuncture** Stimulation type: Bilateral ear seeds with adhesive tape; pressing stimulation was applied until feeling mild discomfort or tingling Points: Shenmen, sympathetic, cardia, stomach, digestive subcortex Treatment start: (No data available) Regime and duration: Points were maintained for 7 days; patients were instructed to stimulate the points three times a day for at least three periods of 3 min duration each day or as soon as they felt nausea. **Antiemetic therapy:** Ondasetron, granisetron, or dexamethasone	**Sham intervention** Stimulation of acupoints not related to CINV (external knee point, vision, shoulder joint, and eye) **Antiemetic therapy**

**Table 3 cancers-16-03197-t003:** Risk of bias of individual studies.

**Author** **Year**	**Random Sequence Generation**	**Allocation Concealment**	**Blinding of Participants**	**Blinding of Personnel**	**Blinding (Outcome Assessment)**	**Incomplete Outcome Data**	**Selective Reporting**
**Altuntaş 2022** [10]	HIGH	UNCLEAR	LOW	HIGH	LOW	LOW	LOW
**Bai** **2024** [11]	LOW	UNCLEAR	HIGH	HIGH	UNCLEAR	LOW	UNCLEAR
**Bintoro** **2022** [12]	LOW	UNCLEAR	LOW	HIGH	LOW	UNCLEAR	UNCLEAR
**Dupuis** **2018** [13]	LOW	UNCLEAR	LOW	HIGH	LOW	LOW	LOW
**Ghezelbasch** **2017** [12]	LOW	UNCLEAR	LOW	HIGH	UNCLEAR	LOW	UNCLEAR
**Gottschling** **2008** [15]	LOW	LOW	HIGH	HIGH	UNCLEAR	LOW	UNCLEAR
**Jones** **2008** [16]	UNCLEAR	UNCLEAR	LOW	HIGH	UNCLEAR	HIGH	UNCLEAR
**Liu** **2018** [17]	UNCLEAR	UNCLEAR	HIGH	HIGH	UNCLEAR	LOW	UNCLEAR
**Varejão** **2019** [18]	UNCLEAR	UNCLEAR	LOW	HIGH	LOW	UNCLEAR	UNCLEAR
**Xie** **2016** [19]	LOW	UNCLEAR	HIGH	HIGH	UNCLEAR	LOW	UNCLEAR
**Yeh** **2012** [20]	LOW	UNCLEAR	LOW	HIGH	LOW	HIGH	UNCLEAR

**Table 4 cancers-16-03197-t004:** Summary of findings.

**Certainty Assessment**							**№ of Patients**		**Effect**		**Certainty**	**Importance**
**№ of Studies**	**Study Design**	**Risk of Bias**	**Inconsistency**	**Indirectness**	**Imprecision**	**Other Considerations**	**Acupuncture**	**Sham**	**Relative** **(95% CI)**	**Absolute** **(95% CI)**		
**Nausea after the intervention**												
4	randomized trials	very serious ^a^	not serious	not serious	not serious	none	122	122	-	SMD **0.57 lower** (0.83 lower to 0.31 lower)	⨁⨁◯◯ Low	CRITICAL
**Vomiting episodes**												
3	randomized trials	very serious ^a^	not serious	not serious	not serious	none	62	62	-	MD **0.16 lower** (0.35 lower to 0.03 higher)	⨁⨁◯◯ Low	CRITICAL

confidence interval; MD: mean difference; SMD: standardized mean difference. ^a^ High risk of bias in randomization of one trial (two intervention arms) and unclear allocation concealment.

## Data Availability

The original data presented in the study are openly available in https://doi.org/10.6084/m9.figshare.26650681 (access on 17 September 2024).

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
