# Peer review of "Efficacy and Safety of Acupuncture and Related Techniques in the Management of Oncological Children and Adolescent Patients: A Systematic Review"

_cancers, 2024, doi:10.3390/cancers16183197_

Round 1

Reviewer 1 Report

Comments and Suggestions for Authors

This paper provided a systematic study to assess the efficacy and safety of acupuncture in alleviating symptoms of pediatric oncology patients. The databases used for statistical analysis are fully included to obtain the credible conclusion. Overally, this invesitigation work is detailed and include many useful information about the potential benefits of acupuncture in alleviating the side effects in pediatric oncology patients. In my impression, I prefer to recommend its acceptance in its present form. One point to note is that the key results are not easily seen in the text and can be summarized with specifc points. For easier readability, a simple Figure including the types of symptoms of pediatric oncology patients alleviated by acupuncture can be also made if possible. 

Author Response

Comment 1: One point to note is that the key results are not easily seen in the text and can be summarized with specifc points

Response: Thank you for your beedback. We belive that the results are clear in the conclusion section "Acupuncture shows promising results as a complementary treatment for reducing nausea and vomiting in pediatric oncology. However, current evidence is insufficient to draw conclusions for other outcomes, such as the number of vomiting episodes, reduction in antiemetic medication use, or fatigue". A "Key results" section could be interesting but this section does not appear in the predefined structure of this journal articles.

Reviewer 2 Report

Comments and Suggestions for Authors

Esther Martínez García, LAc et al. they present some interesting analyzing data about the evidence on the efficacy and safety of acupuncture and its related techniques compared with no treatment s in symptoms associated with cancer and toxicities secondary to cancer treatment in pediatric oncology patients. The manuscript should be well-structured. The topic is relevant and exciting to the field of the journal if some points can be improved:

The explanations about the current treatment methods and effects to nausea and vomiting in young cancer patients are necessary in Introduction, and the simplified explanation of the principles of acupuncture would be better to understand. So, the Introduction should be redesigned.

A brief introduction to the acupuncture points is needed and the most used points in studies should be explained and discussed.

The risk of publication bias should be explained in Materials and Methods.

Comments on the Quality of English Language

Esther Martínez García, LAc et al. they present some interesting analyzing data about the evidence on the efficacy and safety of acupuncture and its related techniques compared with no treatment s in symptoms associated with cancer and toxicities secondary to cancer treatment in pediatric oncology patients. The manuscript should be well-structured. The topic is relevant and exciting to the field of the journal if some points can be improved:

The explanations about the current treatment methods and effects to nausea and vomiting in young cancer patients are necessary in Introduction, and the simplified explanation of the principles of acupuncture would be better to understand. So, the Introduction should be redesigned.

A brief introduction to the acupuncture points is needed and the most used points in studies should be explained and discussed.

The risk of publication bias should be explained in Materials and Methods.

Author Response

Commnet 1: The explanations about the current treatment methods and effects to nausea and vomiting in young cancer patients are necessary in Introduction

Response 1: Thank you for your insightful comment. While we acknowledge that nausea and vomiting are common symptoms in young cancer patients and were frequently addressed in our findings, our review aims to cover the broader applications of acupuncture for oncologic patients, rather than focusing on one specific symptom. Therefore, we believe that expanding on the current treatment methods for nausea and vomiting in the Introduction would narrow the scope of the review and detract from the comprehensive nature of the study.

Comment 2: The simplified explanation of the principles of acupuncture would be better to understand. So, the Introduction should be redesigned.

Response 2:  Thank you for your thoughtful feedback. We understand the importance of presenting a clear explanation of the principles of acupuncture. However, we believe that the current approach in the Introduction provides an appropriate balance between simplicity and the necessary depth required for readers familiar with the subject matter. Our intention was to offer enough detail to ensure scientific accuracy without oversimplifying the complex mechanisms involved. We feel that this approach aligns with the scope and audience of the journal.

Comment 3: A brief introduction to the acupuncture points is needed and the most used points in studies should be explained and discussed.

Response 3: Thank you for your valuable comment. We agree that acupuncture points (acupoints) are highly relevant to understanding the studies included in our review, which is why we have dedicated a section in the Results to describe the acupoints used in the included studies. However, as with our response to Comment 1, this review aims to cover the broader applications of acupuncture across various oncologic conditions, rather than focusing on a single application or symptom. Given this wide scope, it would be impractical to provide a detailed discussion on all possible acupuncture points used for every potential application.

Comment 4: The risk of publication bias should be explained in Materials and Methods

Response 4: Thank you for your valuable comment. In response, we have expanded the explanation in the Methods section to clarify how we intended to assess publication bias and why it was not feasible in this case. The revised section now reads:

"2.4. Publication bias: We initially planned to assess the risk of publication bias by generating a funnel plot, which would allow for the visual detection of asymmetries in the distribution of effect sizes. This method is generally recommended when the meta-analysis includes at least 10 studies, as fewer studies may lead to unreliable or misleading interpretations. However, due to the small number of studies included in our analysis, it was not possible to construct a meaningful funnel plot, and as such, a formal assessment of publication bias could not be performed."

Reviewer 3 Report

Comments and Suggestions for Authors

This review explores the efficacy and safety of acupuncture and related techniques in managing symptoms associated with cancer treatments in pediatric oncology patients. The authors have conducted a comprehensive literature search, incorporating data from a wide range of studies across different countries. The review offers a detailed analysis of various acupuncture techniques, including acupressure, filiform needle, laser acupuncture, and ear acupuncture. It provides valuable insights into the potential benefits of acupuncture in alleviating chemotherapy-induced nausea and vomiting (CIN) in children, highlighting some promising results, particularly in reducing nausea severity. However, the review also emphasizes the need for more high-quality research to substantiate these findings and better understand the clinical relevance of acupuncture in this vulnerable population.

1. While some studies show statistical benefits in reducing nausea severity, others fail to demonstrate significant improvements, particularly in reducing the number of vomiting episodes. This inconsistency may undermine the overall conclusions of the review.

2. Several of the included studies did not provide sufficient data, such as mean differences or confidence intervals, relying instead on p-values to indicate significance.

3. The review identifies a high or unclear risk of bias in many of the included studies, particularly regarding allocation concealment, blinding, and incomplete outcome data. This introduces uncertainty into the review’s findings and suggests that the results should be interpreted with caution.

4.  While acupuncture techniques were generally well-tolerated, with no serious adverse events, it also notes some minor and transient adverse effects, such as local pain, discomfort from tight bands, and itching from ear tapes. These adverse effects, although minor, should be carefully considered in the clinical application of acupuncture in pediatric oncology.

Author Response

Comment 1: While some studies show statistical benefits in reducing nausea severity, others fail to demonstrate significant improvements, particularly in reducing the number of vomiting episodes. This inconsistency may undermine the overall conclusions of the review.

Response: Thank you for your thoughtful comment. We acknowledge the variability in outcomes across the studies, particularly regarding nausea severity and the number of vomiting episodes. We have already addressed this inconsistency in the Conclusions section, noting that the differing results between these two outcomes warrant cautious interpretation of the overall findings. Additionally, we would like to emphasize that nausea severity and the number of vomiting episodes, while related, are distinct outcomes, and it is possible for acupuncture to have varying effects on each.

Comment 2: Several of the included studies did not provide sufficient data, such as mean differences or confidence intervals, relying instead on p-values to indicate significance.

Response: Thank you for your observation. We have already addressed this limitation in the manuscript, particularly in the discussion section

Comment 3: The review identifies a high or unclear risk of bias in many of the included studies, particularly regarding allocation concealment, blinding, and incomplete outcome data. This introduces uncertainty into the review’s findings and suggests that the results should be interpreted with caution.

Response 3: 

We agree with your assessment, and as you noted, we have expressed caution in the interpretation of our findings. Specifically, we state:

"Based on current evidence, acupuncture may provide some benefits in managing chemotherapy-induced nausea (CIN) related to oncological treatments in children when compared to a sham intervention. However, its impact on other outcomes, such as reducing vomiting episodes, alleviating fatigue, and decreasing the need for rescue medication, remains unclear. The heterogeneity, as well as the methodological and reporting limitations of the existing trials, highlight the need for further high-quality research in this field."

Comment 4: While acupuncture techniques were generally well-tolerated, with no serious adverse events, it also notes some minor and transient adverse effects, such as local pain, discomfort from tight bands, and itching from ear tapes. These adverse effects, although minor, should be carefully considered in the clinical application of acupuncture in pediatric oncology.

Response 4: Thank you for highlighting this point. We have already addressed these minor but noteworthy adverse effects in the Results section. We have explicitly noted that acupuncture was generally well-tolerated with no serious adverse events, while also describing the minor and transient adverse effects reported in the studies. We agree that these effects should be considered when evaluating acupuncture for pediatric oncology patients, so whe have added this in the discussion.

Reviewer 4 Report

Comments and Suggestions for Authors

This is a typical systematic review. A plus in this paper is the differentiation between the overall average benefit of acupuncture and the description of the efficacy of the individual interventions as well as the compilation in the tables.

The Introduction should be improved by general descriptions of beneficent therapeutic outcomes due to acupuncture.

Minor points: Please explain TENS in Materials and Methods

What are the symbols O and O with + inside in table 4?

Author Response

Comment 1: The Introduction should be improved by general descriptions of beneficent therapeutic outcomes due to acupuncture.

Response 1: Thank you for your valuable feedback. In response, we have revised the Introduction section to include a broader discussion of the beneficial therapeutic outcomes associated with acupuncture. This expansion provides a more comprehensive overview of how acupuncture can positively impact patient care, particularly in oncology. 

"Acupuncture is a medical technique widely used in leading cancer centers around the world as a complementary treatment in oncology because it does not increase polypharmacy, presents a large margin of safety in expert hands [2–4] and does not interact with the patient's usual treatment. The recently published ASCO guidelines suggest acupuncture for the management of cancer and musculoskeletal pain [5]. It has shown efficacy for cancer pain, or arthralgias secondary to aromatase inhibitors [6]. Oncological acupuncture offers potential benefits for managing symptoms and adverse effects related to cancer and its treatments. For instance, it has been shown to provide relief from chemotherapy-induced nausea and vomiting, reduce pain, and improve overall quality of life for cancer patients. These benefits are especially significant given that current standard treatments may not always be fully effective or could lead to increased polypharmacy and associated toxicity[7]"

Comment 2: Please explain TENS in Materials and Methods

Response 2: Thank you for pointing this out. We have added an explanation of Transcutaneous Electrical Nerve Stimulation (TENS) to the Materials and Methods section. The revised text now reads:

"..., electroacupuncture, transcutaneous electrical nerve stimulation (TENS), moxibustion, ..."

This addition provides clarity on the different modalities of acupuncture and related techniques discussed in the review.

Comment 3: What are the symbols O and O with + inside in table 4?

Response 3: Thank you for your query. The symbols "O" and "O with +" in Table 4 represent the GRADE system for grading the certainty of evidence. Specifically, "O" denotes the level of certainty, with one "+" indicating the lowest certainty and "++++" representing the highest certainty.